# Method for Diagnosing Bearing Faults in Electromechanical Equipment Based on Improved Prototypical Networks

**DOI:** 10.3390/s23094485

**Published:** 2023-05-04

**Authors:** Zilong Wang, Honghai Shen, Wenzhuo Xiong, Xueming Zhang, Jinghua Hou

**Affiliations:** 1Key Laboratory of Airborne Optical Imaging and Measurement, Changchun Institute of Optics, Fine Mechanics and Physics, Chinese Academy of Sciences, Changchun 130033, China; 2Changchun Institute of Optics, Fine Mechanics and Physics, Chinese Academy of Sciences, Changchun 130033, China; 3University of Chinese Academy of Sciences, Beijing 100039, China; 4Jiuquan Satellite Launch Centre, Jiuquan 732750, China

**Keywords:** fault diagnosis, meta-learning, Kullback–Leibler divergence, Gramian angular field

## Abstract

Due to the complexity of electromechanical equipment and the difficulties in obtaining large-scale health monitoring datasets, as well as the long-tailed distribution of data, existing methods ignore certain characteristics of health monitoring data. In order to solve these problems, this paper proposes a method for the fault diagnosis of rolling bearings in electromechanical equipment based on an improved prototypical network—the weight prototypical networks (WPorNet). The main contributions of this paper are as follows: (1) the prototypical networks, which perform well on small-sample classification tasks, were improved by calculating the different levels of influence of support sample distributions in order to achieve the prototypical calculation. The change in sample influence was calculated using the Kullback–Leibler divergence of the sample distribution. The influence change in a specific sample can be measured by assessing how much the distribution changes in the absence of that sample; and (2) The Gramian Angular Field (GAF) algorithm was used to transform one-dimensional time series into two-dimensional vibration images, which greatly improved the application effect of the 2D convolutional neural network (CNN). Through experiments on MAFAULDA and CWRU bearing datasets, it was shown that this network effectively solves the shortcomings of a small number of valid samples and a long-tail distribution in health monitoring data, it enhances the dependency between the samples and the global data, it improves the model’s feature extraction ability, and it enhances the accuracy of model classification. Compared with the prototypical network, the improved network model increased the performance of the 2-way 10-shot, 2-way 20-shot, and 2-way 50-shot classification tasks by 5.23%, 5.74%, and 4.37%, respectively, and it increased the performance of the 4-way 10-shot, 4-way 20-shot, and 4-way 50-shot classification tasks by 12.02%, 10.47%, and 4.66%, respectively. Experimental results show that the improved prototypical network model has higher sample classification accuracy and stronger anti-interference ability compared with traditional small-sample classification models.

## 1. Introduction

Currently, electromechanical equipment plays a crucial role in various industries, from manufacturing to healthcare. It is essential to ensure their regular operation and lifespan to ensure operational continuity and to avoid costly inoperative periods [1]. Prognostics health management (PHM) [2,3] and Remaining Useful Life (RUL) [4] are processes that involve collecting and analyzing data from equipment to detect potential faults and take preventive measures. The PHM process typically involves data collection, processing and preparation, feature selection, and model construction. Data can be collected from various sources, including sensors, telemetry systems, and manual inspection. Researchers can use machine learning algorithms such as regression, decision trees, and neural networks to analyze the data and build models for predicting faults. PHM research in industry is extensive, and this article mainly focuses on the application of PHM in diagnosing failures of rotating bearings in electromechanical equipment. Accurately locating the cause of failures in electromechanical equipment is often difficult, as although the forms of failure may appear simple, the underlying causes are relatively complex. According to the statistics, up to 45% of motor failures are caused by rolling bearings, thus making rolling bearing failures one of the main failures of rotating motors [5].

Traditional PHM research methods are based on physical and mathematical models which require complex modeling, a large number of parameters, and a wealth of expert experience. These methods are primarily used in systems with simple and clear physical models. It is challenging to implement these methods given that modern equipment operates under relatively complex conditions, and thus, they need better portability. For example, Kirubarajan [6] et al. predicted the RUL of the shaft using Forman’s crack growth law; Shen et al. [7] proposed a virtual simulation-based prediction method for the typical failure mode of gear pumps; and Chen [8] et al. established a dynamic system model using two different depths of tooth crack in a sun gear and internal gear, extracting faults via fast spectral kurtosis (FSK).

In data-driven PHM applications, traditional signal processing methods use shallow models, and it is not easy to extract the implicit relationships between different data features in the input dataset; therefore, the diagnostic effect could be better. Yin et al. [9] used SVM for fault diagnosis, Zhao [10] et al. used the Hidden Markov Model (HMM) to diagnose equipment, and Zhang [11] et al. used the Back Propagation Neural Network (BPNN) for equipment fault diagnosis. Deep learning [12,13,14] is a learning method that establishes multi-level representations, converts data features into more abstract modules, learns layer by layer by combining simple, but non-linear, modules, and it shows representations of each layer [15,16]. High-dimensional non-linear features can be transformed into low-dimensional features by using enough hidden layers, thus effectively capturing the hidden information in the data and achieving very complex learning functions [17]. Eren et al. [18] used a convolutional neural network (CNN) classifier for bearing fault diagnoses; Zhang et al. [19] proposed a fault diagnosis method that combines time–frequency feature oversampling (TFFO) with CNN; Yan [20] proposed a fault diagnosis model, MTF-ResNet, based on the Markov transition field and deep residual network; Liu et al. [21] proposed a way of using RNN as an autoencoder for bearing fault diagnosis; Gao et al. [22] optimized fault feature extraction by improving the WDCNN network; and Chen et al. [23] proposed a novel Wide Residual Relation Network (WRRN) for RM intelligent fault diagnoses.

Due to the long-term operation of mechanical equipment under complex working conditions, most of the data collected by the monitoring system is repetitive, and there are few samples consisting of effective and marked fault characteristic information, thus making it difficult to provide a large amount of sample training data for neural networks to train; this causes great difficulties during the application of neural network models. In recent years, the use of small-sample data classification methods when solving the problem of mechanical and electrical equipment fault diagnosis has attracted the attention of many researchers. Xiao et al. [24] used Domain Adaptive (DA) technology and deep transfer learning for fault diagnoses, thus improving performance through knowledge transfer. Chen et al. [25] proposed an adaptive CNN network diagnostic model. Yue et al. [26] extracted rich features using a multi-scale wavelet convolution module and then performed fault diagnosis using a meta-learning module. Li et al. [27] improved the MAML algorithm for a small sample bearing a fault diagnosis. Based on meta-learning, small-sample classification methods have gradually gained the attention of researchers due to their efficient and accurate advantages.

Prototypical network [28] is a meta-learning-based method for small-sample learning; it has performed best when dealing with small-sample classification problems among various models. It uses a support set, S, to extract prototypical vectors for each category, and it classifies query vectors inputted into the query set based on their distance from the prototypical vectors. However, due to the diverse types of faults in mechanical and electrical equipment, such as complex operating conditions, strong time-varying characteristics, and strong long-tail effects in data, the prototypical network structure is prone to ignoring the capturing of feature data, thus making it difficult to quickly capture signal features. This seriously affects its application effectiveness in mechanical and electrical equipment fault diagnosis [29,30].

In this paper, an improved prototypical network model, WPortNet, is proposed and compared with other machine learning models using the same rolling bearing dataset in a comparative experiment, yielding satisfactory results. Firstly, the model uses the Gramian Angular Field (GAF) algorithm to transform one-dimensional time series into two-dimensional vibration images, thus effectively improving the feature extraction effect of the 2D CNN network. Then, classification is performed by calculating the differences between the levels of influence of the supporting sample distribution, and the influence of specific samples can be calculated using the KL divergence in the absence of the sample. The network effectively solves the shortcomings of a small number of effective samples and long-tail distribution in health monitoring data, it enhances the dependence relationship between samples and global data, improves the model’s feature extraction ability, and enhances the accuracy of model classification. Through experiments, it has been demonstrated that the improved prototypical network model has a higher sample classification accuracy and a high level of robustness. The main contributions of this paper are summarized as follows:(1)The prototypical network, which performs well on small-sample classification tasks, was improved by calculating the differences between the influence of the support sample distributions in order to achieve the prototypical calculation. The change in sample influence was calculated using the Kullback–Leibler divergence of the sample distribution. The influence change of a specific sample can be measured by assessing how much the distribution changes in the absence of that sample.(2)The Gramian Angular Field algorithm was used to transform a one-dimensional time series into two-dimensional vibration images, thus greatly improving the application effect of the 2D convolutional neural network.

The rest of the paper is organized as follows: preliminary knowledge is introduced in section two; in section three, the structural design of the proposed model is introduced; some experiments are carried out, and the experimental results are analyzed to evaluate our method against other methods in section four; section five presents the results of the experiments; and finally, we draw conclusions in section six.

## 2. Preliminary Knowledge

Machine learning and prototypical networks have been widely and successfully applied to image classification tasks; however, due to the complex operating conditions, the various types of faults in electromechanical equipment, substantial time variability, and complex frequency components of vibration signals, the prototypical network structure is prone to ignoring the internal correlation of the spectrum. It is severely affected by a large amount of noise, which seriously interferes with the fault characteristics of the signal. In this section, we propose a method using KL divergence to create a new prototypical network, WPorNet, which utilizes the ability of the prototypical network, in terms of image classification, for fault diagnoses of electromechanical equipment.

### 2.1. Meta Learning and Prototypical Networks

Meta Learning, also known as learning to learn, is a crucial approach in machine learning. Its goal is to adapt to new, unseen tasks by learning general learning strategies from a large number of tasks.

Meta learning can be divided into two categories: model-based meta learning and data-based meta learning. Model-based meta learning refers to the learning algorithm, which is a learnable model that can be trained to produce different models for different learning tasks. Common model-based meta-learning algorithms include Model-Agnostic Meta-Learning (MAML) [31] and Reptile [32]. Data-based meta-learning refers to model learning from the training data of multiple tasks so that connections can be found between them and so that it is possible to quickly adapt to new ones. Representative algorithms include prototypical networks, matching networks [33], and relation networks [34].

Prototypical networks are a type of data-driven meta-learning method that achieves data classification tasks by learning prototypical networks from the sample data, and it has the advantage of being simple and efficient when handling small-sample classification problems. It assumes that the samples of each category are clustered around a featured center, and it maps the input onto an embedding vector via a neural network. By calculating the average embedding vector of all support instances in the support set category, the obtained vector is used as the prototypical center of the category, which then realizes the feature extraction of the category; this center is also a parameter of network learning. Then, classification is achieved by measuring the distance between the sample and the center.

A prototypical network divides the training dataset into two parts: the support set and the query set. The support set contains a dataset with K categories and a total of N labeled samples $S=\{(x_{1},y_{1}),(x_{2},y_{2}),\cdot \cdot \cdot ,(x_{N},y_{N})\}$, where $x_{i}\in R^{D}$ is a D-dimensional feature vector and $y_{i}\in \{1,2,\cdot \cdot \cdot ,K\}$ is a sample label. The query set contains H labeled data. First, the data in each category in the support set is inputted into the feature extractor of the prototypical network to learn the nonlinear mapping relationship between the support set and the metric space, and to obtain the M-dimensional feature embedding of each sample in the metric space. $S_{k}$ is a sample set from category k. Then, the feature representation of each category’s health status in the metric space is obtained using the average value of each class’s sample feature embedding, as follows:(1)$$c_{k}=\frac{1}{\left|S_{k}\right|}\sum_{(x_{i},y_{i})\in S_{k}}f_{\phi }(x_{i})$$
where $f_{\phi }(x_{i})$ is the feature embedding function of vector $x_{i}$, which represents the feature of vector $x_{i}$, then, the data of query set Q is inputted to obtain the M-dimensional feature embedding of $c\in R^{M}$. The Euclidean distance $d_{k}$ between c and K prototypicals is calculated and then transformed into the probability of each class using Softmax, represented as:(2)$$p_{\phi }(y=k|\hat{x})=\frac{\exp(-d(f_{\phi }(\hat{x}),c_{k}))}{\sum_{k}\exp(-d(f_{\phi }(\hat{x}),c_{k}))}$$

### 2.2. Gramian Angular Field Transformation

Gramian Angular Field (GAF) can be represented by a Gramian matrix. By using matrix values as pixels in an image, time series can be encoded as images. Gramian Angular Field Transformation is shown in Figure 1.

Wang [35] proposed a method for transforming one-dimensional time series into two-dimensional images using the GAF algorithm. When given a time series, $A=\{a_{1},a_{2},\cdot \cdot \cdot ,a_{n}\}$ with n observations, first, we normalized A so that all values in time series A are within the range of [−1,1] or [0,1]:(3)$${\tilde{a}}_{-1}=\frac{\left(a_{i}-\max(A)+(a_{i}-\min(A))\right)}{\max(A)-\min(A)}$$
(4)$$\mathrm{or}\;{\tilde{a}}_{0}=\frac{a_{i}-\min(A)}{\max(A)-\min(A)}$$

Next, the time series was transformed into polar coordinates:(5)$$\phi_{i}=\arccos({\tilde{a}}_{i}),-1\leq {\tilde{a}}_{i}\leq 1,{\tilde{a}}_{i}\in \tilde{A}\;r_{i}=\frac{t_{i}}{N},t_{i}\in N$$
where $t_{i}$ is the timestamp and N is a constant factor that is used to adjust the range of the polar coordinate system. The angle value $\phi_{i}$ in the polar coordinate system is the inverse cosine of the time series ${\tilde{a}}_{i}$, and the radius in the polar coordinate system is the ratio of the timestamp to the constant factor. This achieves a transformation from the amplitude variation of the time series to the angle variation in the polar coordinate image. Then, the Gramian Field is generated by defining the sum of the trigonometric functions of each point in the interval:(6)$$GAF=\left(\begin{array}{ccc} \cos(\phi_{1}+\phi_{1}) & \ldots & \cos(\phi_{1}+\phi_{N})\\ \vdots & \ddots & \vdots \\ \cos(\phi_{N}+\phi_{1}) & \cdots & \cos(\phi_{N}+\phi_{N}) \end{array}\right)$$

It is a bijection, and when ϕ∈[0,π], cos(ϕ) is monotonically increasing. The value of GAF not only depends on the time stamp interval but also on the absolute position of the time series.

## 3. Fault Diagnosis Method Based on WproNet

We calculated the distributional influence of the support-set samples and incorporated it into the network for reevaluation. The differences between the levels of distributional influence among samples can be measured by how much the distribution changes when the sample is removed, unlike in the prototypical network, where all samples are treated equally. This enhances the dependence between the samples and the global data, thus effectively addressing the long-tail distribution characteristics of health monitoring data and significantly improving the accuracy of model classification.

### 3.1. Improved Model Architecture

The model is mainly composed of a preprocessing layer, encoder layer, and distribution-prototypical layer. The raw signal is transformed by GAF to obtain the vibration image in the preprocessing layer. Then, for each category, the vibration image is divided into support and query sets. In the encoder layer, a four-layer CNN embedding function $f_{\phi }$ is used to extract features and generate a prototypical representation for each category. After the features extracted from the two channels are passed through a convolutional block, the distributional weight of each support sample is calculated to obtain the new probability of the category. Next, the distribution–prototypical layer maps onto the feature space to calculate the distributional distance between the sample and the prototypical layer, and the probability of the sample in each category is calculated using Softmax. The model’s structure is shown in Figure 2.

### 3.2. Encoder Layer

WproNet uses a four-layer convolutional neural network (CNN) as its encoder. The structure of the encoder is shown in Figure 3. Each convolutional block is composed of four identical encoder modules, each of which contains a convolutional layer (Conv2D), a batch normalization layer (BatchNorm2D), a linear rectification function layer (ReLU), and a maximum pooling layer (MaxPool2D). Finally, the Flatten layer is introduced to flatten the feature vectors of each sample into a one-dimensional vector input for the loss function. The network automatically learns the features of the signals from the grayscale vibration images to train the embedding function $f_{\phi }$, and it then prepares for the next operation.

The convolutional layer extracts features by sliding a convolution kernel over an input vector with a certain window size. When performing convolution, the width of the convolution kernel should be the same as the dimension of the embedding vector output of the embedding layer. Here, a convolution kernel of size m is used for the convolution operation, and the hidden representation of each feature in the image is obtained via calculations.
(7)$$h_{i}=CNN(s_{i-\frac{m-1}{2}},\cdot \cdot \cdot ,s_{i+\frac{m-1}{2}})$$
where CNN() represents the convolutional operation and the image after undergoing convolutional operation can be represented as:(8)$$x=(h_{1},h_{2}\cdot \cdot \cdot ,h_{n})$$

After the convolution operation, a ReLU activation function is added:(9)ReLU=max(0,a)

### 3.3. Distribution–Prototypical Layer

#### 3.3.1. K–L Divergence

The Kullback–Leibler Divergence (KL divergence), also known as relative entropy, is generally used to measure the similarity or degree of difference between two probability distribution functions. Given the two distribution functions, P(X) and Q(X), their KL divergence is defined as follows:(10)$$D_{KL}\left[P(X)||Q(X)\right]=\sum_{x\in X}\left[P(x)\log\frac{P(x)}{Q(x)}\right]=E_{x\sim P(x)}\left[\log\frac{P(x)}{Q(x)}\right]$$

Thus, given the two datasets with distributions A and, respectively, their KL divergence is:(11)$$D_{KL}\left[A||B\right]=\sum_{i}\left[A\log\frac{A}{B}\right]$$
where ϕ represents the function that maps onto the latent space. When KL divergence is equal to 0, A=B indicates that the distribution of A and B is highly consistent. Samples with a high KL divergence represent deviations from the distribution.

#### 3.3.2. Distribution–Prototypical Layer Design

Unlike the prototypical network, after mapping the samples onto the feature space using the feature extractor, we used the K–L divergence to measure the consistency between the test samples and their corresponding dataset distribution. The weight of the sample can be measured by how much the distribution changes when the sample is not present in the dataset:(12)$$D_{KL}(x_{i})=D_{KL(}{}_{\phi )}\left[S||S-x_{i}\right]=\sum_{i}\left[S\log\frac{S}{S-x_{i}}\right]$$

We define the sample weight W by normalizing the K–L divergence of the sample, as follows:(13)$$W(f_{\phi }(x_{i}))=D_{KL}({\tilde{x}}_{i})$$

Therefore, the new feature representation of each category is:(14)$$c_{k}=\frac{\sum_{i=1}^{\left|S_{k}\right|}W(f_{\phi }(x_{i}))f_{\phi }(x_{i})}{\sum_{i=1}^{\left|S_{k}\right|}W(f_{\phi }(x_{i}))}$$

Finally, we calculated the loss function:(15)$$J(x,c_{k})=-\log\frac{1}{q}\sum_{i=1}^{q}\frac{\exp(-d(f_{\phi }(\hat{x}),c_{k}))}{\sum_{k}\exp(-d(f_{\phi }(\hat{x}),c_{k}))}$$
where q denotes the number of query instances.

When a loss function is generated, the stochastic gradient descent (SGD) is used to train the network. Through multiple updates, optimal model parameters ϕ and prototypical networks of various health states are obtained. By calculating the sample weights, new probabilities for each class are obtained, and the model’s prediction result is obtained.

### 3.4. Fault Diagnosis Process

The improved prototypical network is applied during the fault diagnosis process concerning mechanical and electrical equipment bearings, which is divided into three parts: data preprocessing, network model training, and model testing.

#### 3.4.1. Dataset

To verify the effectiveness of the proposed method, we conducted experiments using two real-world open datasets from Case Western Reserve University (CWRU) [36] and the Machinery Fault Database (MAFAULDA) [37]. The experimental device is shown in Figure 4.

The CWRU dataset includes one normal condition type and three fault types: (i) inner race fault, (ii) outer race fault, and (iii) ball fault. These correspond with the four load types (0HP, 1HP, 2HP, and 3HP) and three diameter types (0.007″, 0.014″, and 0.021″). The dataset is divided into four categories: 48k Hz baseline, 48k Hz drive-end fault, 12k Hz drive-end fault, and 12k Hz fan-end fault. We only selected the dataset from the 12k Hz drive-end fault (corresponding with a sampling rate of 12 KHz) and from a unidirectional accelerometer installed at the drive-end bearing position in the CWRU dataset.

The MAFAULDA dataset is composed of a time series with five fault conditions—(i) horizontal misalignment, (ii) vertical misalignment, (iii) imbalance, (iv) underhang bearing fault (outer race fault, ball fault, and cage fault), and (v) overhang bearing fault (outer race fault, ball fault, and cage fault). Only the data from the underhang bearing fault, which were collected by three accelerometers on the radial, axial, and tangential directions, were selected here.

#### 3.4.2. Data Preprocessing

The vibration signal of the electromechanical equipment bearing is a one-dimensional time series, and we needed to generate vibration images before use. More specifically, we divided each group of data into groups of 400 data points. Then, we used the Gramian Angular Field (GAF) method to encode the values of the time series as angular cosines and the timestamps as radii in order to represent the time series in polar coordinates. This process can preserve time dependencies and then calculate its GAF image in accordance with the defined GAF formula. This achieves a transformation from time series to image, thus obtaining the vibration images. The vibration signals of various fault types, and the vibration images generated from them, are shown in Figure 5.

#### 3.4.3. Training and Testing of Models

1.The model parameters and various prototypes were trained using support and query set data. During the training process, the stochastic gradient descent (SGD) method was used to optimize and adjust parameters ϕ until a better performance was achieved. The training process of the WproNet is shown in Algorithm 1.

**Algorithm 1** Training process of the WproNet **Input:** Training Dataset $D=\{(x_{1},y_{1}),(x_{2},y_{2}),\cdot \cdot \cdot ,(x_{N},y_{N})\}$
**Output:** The trained network parameter $f_{\phi }$
**Begin:**
1:       **For**
i
**in** [1,N] **do**:
2:           $D_{i}\leftarrow GAF(D_{i})$
3:       **End For**
4:       **For** epoch **to** set value, **do**:
5:           **For** epoch **to** set value, **do**:
6:               **For**
i**in** $\left[1,y_{N}\right]$ **do**:
7:                   Randomly take $N_{s}$ samples $\left\{(x_{1},y_{1}),\cdot \cdot \cdot ,(x_{N_{s}},y_{N_{s}})\right\}$ from D as the support set S
8:                   Randomly take $N_{q}$ samples $\left\{(x_{1},y_{1}),\cdot \cdot \cdot ,(x_{N_{q}},y_{N_{q}})\right\}$ from D/S as the query set Q
9:                   **For**
i**in** $\left[1,N_{s}\right]$ **do**:
10:                      Compute distribution changes of the samples: $D_{KL}(x_{i})=\sum_{i}\left[S\log\frac{S}{S-x_{i}}\right]$
11:                      Normalization as weight: $W(f_{\phi }(x_{i}))\leftarrow D_{KL}({\tilde{x}}_{i})$
12:                      Compute prototypical of the samples: $c_{k}=\frac{\sum_{i=1}^{\left|S_{k}\right|}W(f_{\phi }(x_{i}))f_{\phi }(x_{i})}{\sum_{i=1}^{\left|S_{k}\right|}W(f_{\phi }(x_{i}))}$
13:                  **End For**
14:              **End For**
15:               J←0
16:              **For**
i
**in**$\left[1,N_{s}\right]$**do**:
17:                  **For** $\left[x,y_{j}\right]$
**in**
$Q_{N_{q}}$**do**:
18:                      Update loss: $J\leftarrow J+J(x,c_{k})$
19:                      Update the parameter ϕ via the SGD method
20:                 **End For**
21:             **End For**
22:          **End For**
23:       **End For**

2.After the network model was trained and various prototypes were obtained, the validation set was used to perform fault diagnosis testing. Figure 6 shows the flowchart of our model. The hyperparameters and parameter settings of the WproNet are shown in Table 1 and Table 2.

## 4. Results

### 4.1. Comparative Experiments

In order to verify the outstanding performance of the proposed model, experiments were conducted using the same bearing dataset from the CWRU and MAFAULDA. The experiments were compared with representative classification algorithms based on statistics (SVM), CNNs (WDCNN), meta-learning (matching networks), and the original prototypical network. The settings for the parameters and hyperparameters are shown in Table 1.

SVM [9]: The Support Vector Machine is a supervised learning algorithm for classification and regression analysis. It is a binary classification model that finds the optimal hyperplane to achieve classification. SVM can handle non-linear classification problems well;WDCNN [22]: Deep Convolutional Neural Networks with Wide First-layer Kernels is a traditional machine learning model based on deep convolutional neural networks (DCNNs). Its main feature is that it uses wide convolutional kernels to increase the number of features and reduce network depth. It requires a large number of samples for training;Matching Networks [33]: Matching Networks is a meta-learning method that uses an attention-based approach to compare input samples with samples in the support set, thus enabling rapid model adaptation.Prototypical network [28];DSN—Conv4 [38]: Discriminative Deep Subspace Networks, the backbone of which is composed of Conv4;PNMD [39]: Prototypical network based on the Manhattan distance.

The network model is built using the PyTorch 1.10.2 deep learning framework, and it is accelerated by CUDA 11.3 and CUDNN 8.3.2. The system environment is Windows 10, and the processor is Intel^®^ Core™ i7-9700K CPU @ 3.60GHz, 16.0GB RAM, and the Nvidia GeForce RTX 3090 graphics card.

Accuracy is used as the primary evaluation metric.

We set different parameters for the N-way K-shot task in the experiment, selecting two and four categories and randomly selecting 10/20/50 images from each class to create a support set and ten images were selected for the query set. Training and fine-tuning were conducted for 2-way 10-shot/2-way 20-shot/2-way 50-shot and 4-way 10-shot/4-way 20-shot/4-way 50-shot, respectively, with episodes set to 100. Each epoch contained 100 episodes, and the epochs were set to 1000. We calculated the accuracy using the method of repeating the same scene 20 times and taking the average value. The results are shown in Table 3 and Table 4.

The experimental results show that for small-sample classification tasks, the performance of WProNet is better than traditional machine learning models and meta-learning methods. When different numbers of categories and sample sizes were used, the method proposed in this paper, WProNet, achieved the highest accuracy rate among the five fault diagnosis methods. This is because traditional machine learning models and CNN models have serious overfitting problems in small-sample classification tasks which require a large number of samples for training to achieve satisfactory results. In particular, the WDCNN network has poor accuracy when there are only ten samples. The accuracy rates for the two classification scenarios were only 42.35% and 33.42% with regard to the experiment concerning the CWRU dataset, respectively. With the increase in the number of samples, the overfitting of each network is alleviated, and the accuracy rate is improved. The SVM model shows unexpectedly high accuracy rates, achieving performance comparable to the matching network in various classification experiments. In the comparison experiment with the prototypical network, the improvement in accuracy of the four-way classification task is significantly better than that of the two-way classification task, and the improvement effect is most obvious in the 4-way 10-shot scenario, with an improvement of 12.02%. As the number of categories decreased and the number of samples increased, the gap between the two models gradually narrowed.

#### 4.1.1. Ablation Experiment

We conducted ablation experiments on two datasets to demonstrate the effectiveness of the improved algorithm, as shown in Table 5 and Table 6. These experimental results indicate that the prototypical network was calculated with the added distribution difference weight, which significantly outperformed the original prototypical network. The model with added weights enhanced the dependent relationship between the samples and global data, improved the model’s feature extraction ability, and increased the accuracy of the model classification. All experimental metrics were improved, thus demonstrating the effectiveness of the model.

#### 4.1.2. Training Time Analysis

The training runtime of experimental models and our model, with regard to both datasets, CWRU and MAFAULDA, are shown in Table 7. To ensure a fair comparison, the hyperparameters of all the methods follow their original settings. Our training runtime is nearly the same as that of the match networks’ runtime, and slightly greater than that of ProtoNet and PNMD. It is evident that even among the various models based on the same Conv-4 backbone networks, our model is highly efficient. Our approach achieves the best performance with only a modest increase in total training time.

#### 4.1.3. Visualization Analysis

We used t-Distributed Stochastic Neighbor Embedding (t-SNE) [40] to visualize the performance of WProNet and prototypical networks on the test set, regarding 4-way 50-shot. The results are shown in Figure 7. It is evident that both WProNet and prototypical networks achieve a very good performance, but the prototypical networks’ ability to generalize is much weaker than that of the WProNet.

#### 4.1.4. Comparison of WProNet with Several Other Models

To observe the performance of the WProNet in various categories, we plotted the confusion matrix of a random test result on the validation set with 50 samples, as shown in Figure 8. The vertical axis represents the true label of the validation set, the horizontal axis represents the predicted label of the validation set, and the main diagonal axis represents the number of correctly predicted samples as per the network model.

The WProNet model incorporates the influence of the sample distribution into the network and reevaluates it, instead of treating all samples equally as in traditional prototypical networks. This effectively solves the characteristics of the long-tail distribution of health monitoring data, and it enhances the dependence relationship between samples and global data, thus effectively improving the accuracy of model classification.

## 5. Discussion

When setting the initial learning rate, we tested the prototypical network and our model from 0.0001 to 0.001 in increments of 0.0001. As shown in Figure 9, we found that the best effect was achieved when the initial learning rate was set to 0.0005, and if it was set to other values, the effect would be reduced. Learning rates that are either too small or too large can lead to a decrease in model accuracy.

When selecting the number of CNN layers, we tested 3, 4, 5, and 6 layers separately. The results of the experiment show that only when there are four layers does the model have the highest accuracy, and other situations will lead to a certain degree of underfitting and overfitting, which affects the accuracy of the model. It also shows that simply increasing the number of CNN layers will not increase the performance of the model.

Improving the feature optimization learning ability of machine learning methods is an important means to solve complex mechanical vibration signal problems. The WProNet model, which is based on the prototypical network proposed in this paper, is used to detect mechanical and electrical equipment bearing faults, incorporate the influence of sample distribution into the network, and allocate weights based on the proportion of sample distribution in the dataset, thus effectively solving the defects of health monitoring data regarding long-tail distributions, and enhancing the dependence relationship between samples and global data. By establishing feature prototypical networks of various health states in the measurement space, and then classifying and querying the health status of samples through distance metrics, the overfitting problem can be effectively alleviated, and the accuracy of fault diagnoses in small samples can be improved.

## 6. Conclusions

This article proposes a prototypical network based on the differences between computed sample distributions in order to identify the healthy state of rotating machinery with few samples. To test the performance of the proposed model, we conducted comparative experiments on the bearing dataset at CWRU and MAFAULDA, with other machine learning models such as SVM, WDCNN, and a prototypical network. The improved network model that was used on the CWRU dataset, increased the performance quality by 5.23%, 5.74%, and 4.37% in the 2-way 10-shot, 2-way 20-shot, and 2-way 50-shot classification tasks, respectively. It also increased the performance quality by 12.02%, 10.47%, and 4.66% in the 4-way 10-shot, 4-way 20-shot, and 4-way 50-shot classification tasks, compared with the prototypical network. The results from the MAFAULDA dataset also showed significant improvement.

The experimental results have shown that this method demonstrates great potential for complex signal classification and recognition tasks. It can learn data features from the minimum number of samples, thus indicating that the model is feasible and effective for classifying and recognizing complex signals and time-series data. Future studies will focus on applying this model to other industrial applications to develop real-time fault detection for different types of machines. Additionally, recognizing and denoising complex signals will also be topics of interest for future research.

## Figures and Tables

**Figure 1 sensors-23-04485-f001:**
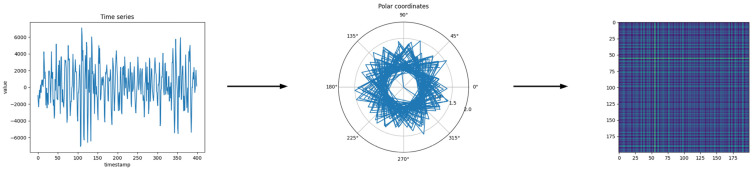
Gramian Angular Field Transformation.

**Figure 2 sensors-23-04485-f002:**
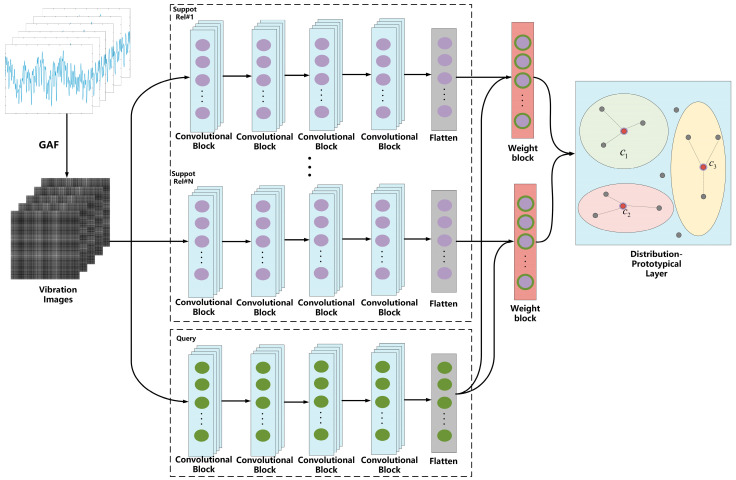
Structure of the Improved Model.

**Figure 3 sensors-23-04485-f003:**
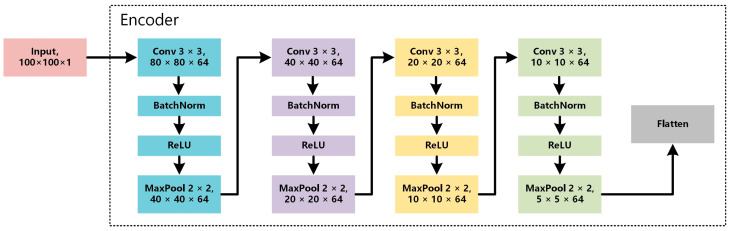
Structure of Encoder.

**Figure 4 sensors-23-04485-f004:**
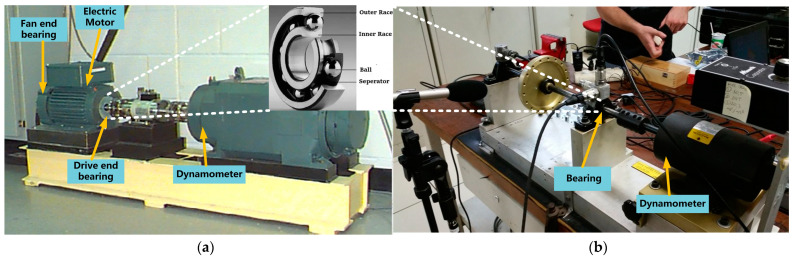
Experimental device. (**a**) CWRU; (**b**) MAFAULDA.

**Figure 5 sensors-23-04485-f005:**
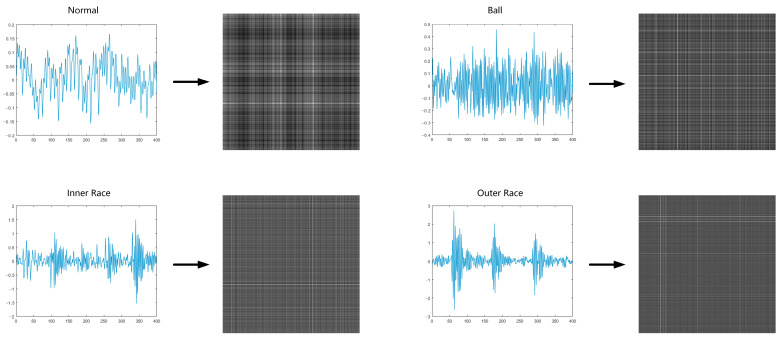
Vibration signals of various fault types and their generated vibration images.

**Figure 6 sensors-23-04485-f006:**
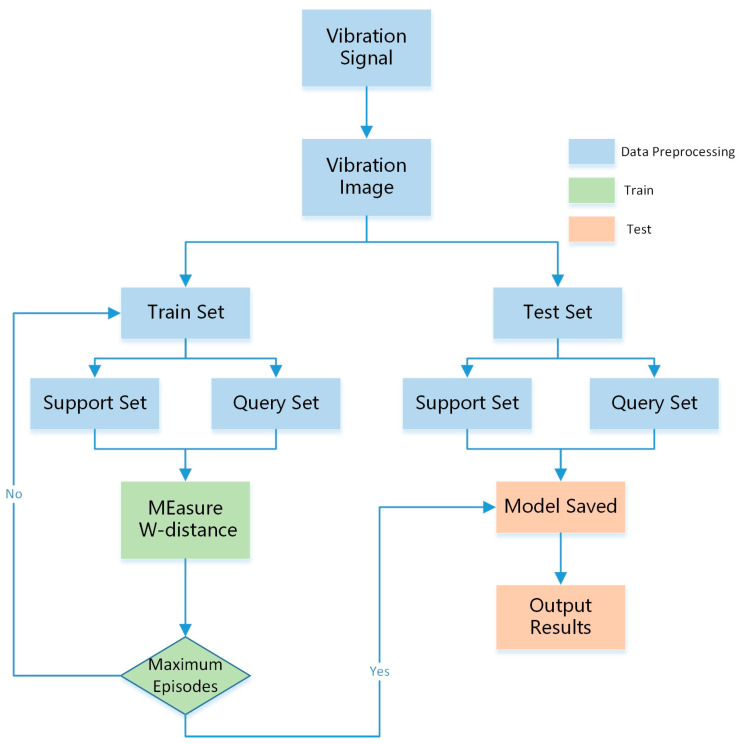
Flowchart of our model.

**Figure 7 sensors-23-04485-f007:**
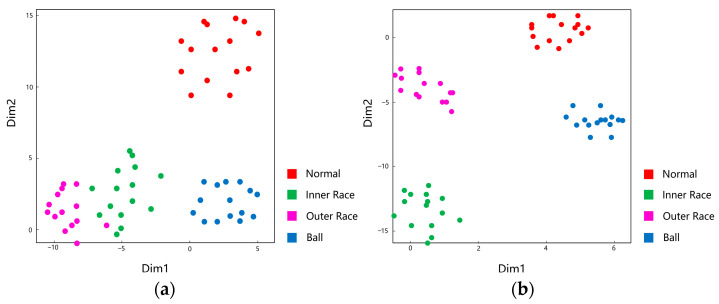
Visualization of the features extracted by the Prototypical Networks and the weight prototypical networks (WProNet) via t-SNE. (**a**) Prototypical Networks; (**b**) WProNet.

**Figure 8 sensors-23-04485-f008:**
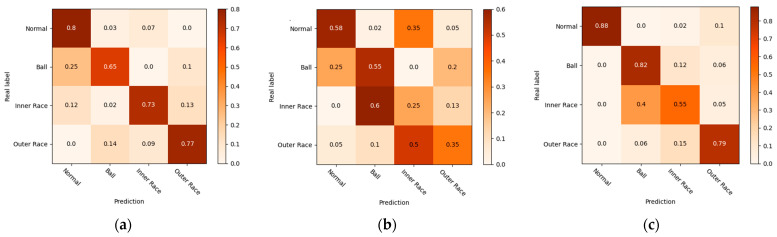
Confusion matrix of Models. (**a**) SVM; (**b**) WDCNN; (**c**) Match Networks; (**d**) Prototypical Networks; (**e**) WProNet (ours).

**Figure 9 sensors-23-04485-f009:**
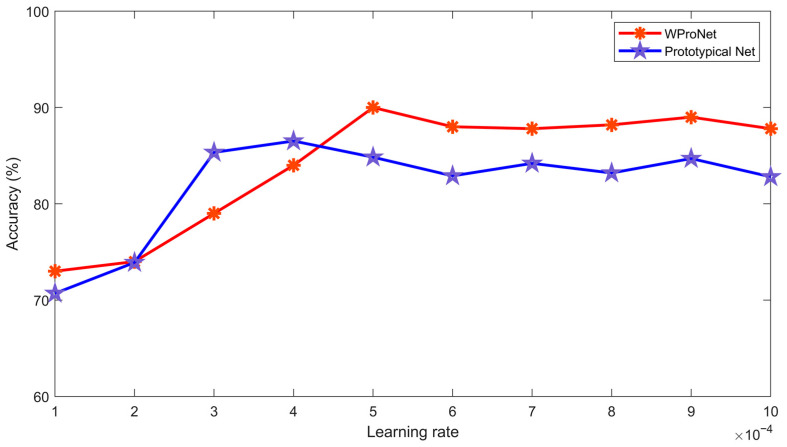
The 0.0001–0.001 interval for the learning rate (0.0001 is the step size) to test the Prototypical Network and our model.

**Table 1 sensors-23-04485-t001:** Hyperparameter settings of the WproNet.

Name	Channels	Kernel Size	Stride	Input Size	Output Size	Activation Function
convolutional block1	64	3 × 3	1 × 1	80 × 80 × 64	40 × 40 × 64	Relu
convolutional block2	64	3 × 3	1 × 1	40 × 40 × 64	20 × 20 × 64	Relu
convolutional block3	64	3 × 3	1 × 1	20 × 20 × 64	10 × 10 × 64	Relu
convolutional block4	64	3 × 3	1 × 1	10 × 10 × 64	5 × 5 × 64	Relu
Distribution-Prototypical Layer	/	/	/	1600	1600	Softmax

**Table 2 sensors-23-04485-t002:** Parameter settings of the WproNet.

Parameter Type	Parameter Value
Optimizer	SGD
Initial learning rate	0.0005
Learning rate decay period	2000 episodes

**Table 3 sensors-23-04485-t003:** Average accuracies of the results for CWRU (%).

Model	2-Way			4-Way		
	10-Shot	20-Shot	50-Shot	10-Shot	20-Shot	50-Shot
SVM	74.78 ± 1.59	77.89 ± 1.41	82.85 ± 1.03	50.67 ± 2.62	58.21 ± 2.87	74.06 ± 1.53
WDCNN	42.35 ± 2.32	47.12 ± 2.17	59.64 ± 1.52	33.42 ± 2.89	38.80 ± 2.21	43.71 ± 1.98
Match Networks	72.93 ± 1.39	74.86 ± 1.31	80.31 ± 0.92	48.56 ± 2.28	57.78 ± 2.49	77.21 ± 1.67
Prototypical Networks	85.14 ± 1.13	85.68 ± 0.96	91.87 ± 0.70	59.77 ± 1.71	67.67 ± 1.82	85.02 ± 1.33
DSN- Conv4	88.39 ± 0.66	89.61 ± 0.53	95.52 ± 0.34	71.08 ± 0.70	72.36 ± 0.54	88.67 ± 0.30
PNMD	85.54 ± 1.61	85.80 ± 1.13	92.27 ± 0.69	61.37 ± 1.85	68.19 ± 1.35	87.05 ± 0.92
WProNet(ours)	**90.37 ± 0.83**	**91.42 ± 0.77**	**96.24 ± 0.75**	**71.79 ± 0.91**	**78.14 ± 1.14**	**89.68 ± 0.96**

**Table 4 sensors-23-04485-t004:** Average accuracies of the results for MAFAULDA (%).

Model	2-Way			4-Way		
	10-Shot	20-Shot	50-Shot	10-Shot	20-Shot	50-Shot
SVM	76.49 ± 1.64	80.16 ± 1.32	83.37 ± 0.88	51.67 ± 2.47	59.04 ± 2.20	77.43 ± 1.43
WDCNN	44.61 ± 2.09	46.12 ± 2.19	61.36 ± 1.87	34.12 ± 3.05	41.06 ± 2.51	47.26 ± 2.14
Match Networks	75.80 ± 1.40	78.20 ± 1.41	83.32 ± 0.85	50.61 ± 2.09	61.88 ± 1.89	82.64 ± 1.55
Prototypical Networks	86.02 ± 1.09	88.64 ± 1.05	93.50 ± 0.65	62.28 ± 1.38	70.07 ± 2.06	88.16 ± 0.83
DSN- Conv4	89.03 ± 0.51	91.15 ± 0.60	94.76 ± 0.38	70.84 ± 0.40	77.76 ± 0.45	90.33 ± 0.41
PNMD	85.79 ± 0.86	88.31 ± 0.96	93.97 ± 0.59	62.16 ± 1.66	70.19 ± 1.39	89.25 ± 0.86
WProNet(ours)	**91.70 ± 0.85**	**93.71 ± 0.70**	**96.45 ± 0.69**	**75.35 ± 1.32**	**80.73 ± 1.16**	**91.93 ± 0.94**

**Table 5 sensors-23-04485-t005:** Comparison of ablated model structures for CWRU (%).

Model	2-Way			4-Way		
	10-Shot	20-Shot	50-Shot	10-Shot	20-Shot	50-Shot
ProNet	85.14 ± 1.13	85.68 ± 0.96	91.87 ± 0.70	59.77 ± 1.71	67.67 ± 1.82	85.02 ± 1.33
W+ProNet	90.37 ± 0.83	91.42 ± 0.77	96.24 ± 0.75	71.79 ± 0.91	78.14 ± 1.14	89.68 ± 0.96

**Table 6 sensors-23-04485-t006:** Comparison of ablated model structures for MAFAULDA (%).

Model	2-Way			4-Way		
	10-Shot	20-Shot	50-Shot	10-Shot	20-Shot	50-Shot
ProNet	86.02 ± 1.09	88.64 ± 1.05	93.50 ± 0.65	62.28 ± 1.38	70.07 ± 2.06	88.16 ± 0.83
W+ProNet	91.70 ± 0.85	93.71 ± 0.70	96.45 ± 0.69	75.35 ± 1.32	80.73 ± 1.16	91.93 ± 0.94

**Table 7 sensors-23-04485-t007:** Training runtime comparison of models on CWRU and MAFAULDA datasets, under 2-way n-shot and 4-way n-shot classification scenarios.

Model	Training Tasks	2-Way			4-Way		
		10-Shot	20-Shot	50-Shot	10-Shot	20-Shot	50-Shot
SVM	100,000	0.5 h	0.8 h	1.2 h	1.3 h	1.8 h	2.3 h
WDCNN	100,000	1 h	1.3 h	2 h	2.6 h	3 h	4.3 h
Match Networks	100,000	1.7 h	2 h	2.2 h	2.5 h	4 h	6 h
Prototypical Networks	100,000	1.5 h	1.8 h	2.4 h	2.5 h	3.8 h	5 h
DSN-Conv4	100,000	8 h	13 h	19 h	15 h	21 h	35 h
PNMD	100,000	1 h	1.2 h	2 h	1.5 h	1.7 h	2.5 h
WProNet(ours)	100,000	2 h	2.3 h	3.5 h	2.4 h	3.3 h	4.4 h

## Data Availability

Not applicable.

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
