# Peer review of "Method for Diagnosing Bearing Faults in Electromechanical Equipment Based on Improved Prototypical Networks"

_sensors, 2023, doi:10.3390/s23094485_

Round 1

Reviewer 1 Report

The innovation is not enough. the content of this paper needed to be further improved. Here is my specific suggestion:

(1) The Abstract should emphasize the problem of the existing methods. And, the idea of this paper should be illustrated to show how to solve these problems. Besides, the logic of the Abstract should be improved.

(2) The contribution of this paper can be summarized in the introduction. And, the framework of this paper can be also added at the end of the introduction.

(3) What is the specific parameter in your proposed neural network such as Kernal size, output channel, and activation function in every layer? Please illustrate the configuration of the proposed neural network by constructing a Table.

(4) The connection between the sub-chapter should be improved. The author should emphasize the procedure of the proposed method. Section 3.1 to Section 3.2 looks isolated.

(5) What is the innovation of this paper? Is the proposed neural network in this paper? Why not emphasize the priority of the proposed neural network? Why do you choose this neural network?

(6) The overall process of the proposed algorithm can be summarized in a Table.

(7) The equation of this paper doesn’t show the process of the proposed method.

(8) The experiment is not enough. The authors should add other datasets to verify the effectiveness of the proposed method.

(10) The ablation experiment should be introduced in this paper to illustrate the effectiveness of the proposed method.

(11) The evaluating index of this paper is single. How long does the author train the neural network? And, the experiment should be repeated many times to avoid randomness. The standard deviation should be introduced to evaluate the robustness of the proposed network.

(12) Please add the classification result by t-SNE to vividly depict the classification result.

(13) The quality of Figure 8 should be improved.

(14) More state of art methods should be compared with the proposed method to verify the performance.

Reviewer 2 Report

In this paper, the authors suggested an improved prototypical network to diagnose the fault of mechanical and electrical equipment bearings, which is divided into three parts: data preprocessing, network model training, and model testing. They also conducted comparative experiments on the bearing dataset with other machine with other learning models such as SVM, WDCNN, and prototypical network in order to verify and compare their proposed model with other models. They concluded that the proposed model improved the performance compared to the prototypical network. In addition, they claimed that the proposed model effectively solves the defect of health monitoring data longtail distribution and enhances the dependence relationship between samples and global data, specifically the accuracy of fault diagnosis under small samples can be improved.

This is a clear, concise, and well-written manuscript. The introduction is relevant and theory based. Sufficient information about the previous study findings is presented for readers to follow the present study rationale and procedures. The text is clear and easy to read, and the results are sufficiently discussed. Overall, the manuscript is well thought out and written, the objectives clearly stated, applied methods are advanced, data statistically analyzed. However, to accept the paper, the authors should consider the next comments:

1. Page 2, Line 49-51: The authors stated, that: "According to statistics, up to 45% of motor failures are caused by rolling bearings, making rolling bearing failures one of the main failures of rotating motors.". Please give some references to confirm your statement.

2. Page 11, Line 347-349: The authors stated that: "The horizontal axis represents the true label of the validation set, the vertical axis represents the predicted label of the validation set, and the main diagonal represents the number of correct samples predicted by the network model.". However, it does not match in the Figure 7. In Figure 7: The horizontal axis represents the predicted label of the validation set, the vertical axis represents the real label of the validation set. Please correct it.

3. In "Abstract" and "Conclusions", the authors stated that: "Compared with the prototypical network, the improved network model has increased the performance of the 2-way 10-shot, 2-way 20-shot, and 2-way 50-shot classification tasks by 5.51%, 5.85%, and 3.84%, respectively, and increased the performance of the 4-way 10-shot, 4-way 20-shot, and 4-way 50-shot classification tasks by 11.98%, 10.32%, and 5.24%, respectively.". However these percentages are not mentioned in the content of the paper.

Round 2

Reviewer 1 Report

Many of my concerns had been addressed properly. The quality of this paper had been improved.

Before acceptance of this paper, the authors should address two problems.

       Q1: In the classification of t-SNE, the name of different fault types should be drawn out in Fig. 7. Please add a legend in Fig. 7.

       Q2: Please add the unit in the Y-axis of Fig. 9.
